# Managers’ Well-Being in the Digital Era: Is it Associated with Perceived Choice Overload and Pressure from Digitalization? An Exploratory Study

**DOI:** 10.3390/ijerph16101746

**Published:** 2019-05-17

**Authors:** Sabrina Zeike, Kyung-Eun Choi, Lara Lindert, Holger Pfaff

**Affiliations:** Institute of Medical Sociology, Health Services Research, and Rehabilitation Science (IMVR), Faculty of Human Sciences, Medical Faculty, University of Cologne, Eupener Strasse 129, 50933 Cologne, Germany; kyung-eun.choi@uk-koeln.de (K.-E.C.); lara.lindert@uk-koeln.de (L.L.); holger.pfaff@uk-koeln.de (H.P.)

**Keywords:** manager, leadership, digitalization, digital transformation, job demands, choice overload, pressure from digitalization, psychological well-being

## Abstract

Due to the current digital transition, companies are under pressure to pursue digitalization and often initiate far-reaching transformation processes. As a result, managers must drive change within a company and are involved in important decision-making processes. In the present study, we focused on two cognitive job demands in managers related to change due to digital transformation: perceived choice overload and pressure from digitalization. We assumed that the extent of challenging cognitive demands at work is rising and negatively influencing managers’ psychological well-being. We conducted an online survey with a sample of 368 upper-level managers from a large ICT-company, where, at the time of the study, extensive transformation processes were taking place. Using multivariate regression analysis, potential prognostic effects on well-being were tested. Results showed that lower well-being was significantly associated with higher choice overload, but not with perceived pressure from digitalization. In our explorative study, we investigated two potential job demands in managers that, to our knowledge, have not yet been scientifically tested. Given the unsettled state of the field, it is important to try to further understand when choice overload and pressure from digitalization occur and when these may trigger negative health consequences.

## 1. Introduction

In recent decades, industrialization and digitalization have substantially changed society and the nature of work [1]. Due to increasing digitalization, companies have faced changes that have led to new opportunities but have also led to diverse challenges such as rapid technological change and increased work complexity [2]. Digitization (i.e., transforming analogue data into digital data) is the framework for digitalization, which can be defined as ‘the exploitation of digital opportunities’ [2]. Digital transformation refers to ‘the process that is used to restructure economies, institutions, and society on a system level’ [2] Digitalization has led to far-reaching transformation processes in all industries and has created enormous pressure to transform existing businesses. According to previous research, digital transformation has fundamental effects on businesses, societies, and individuals [3,4,5,6,7]. Digitalization is supposed to change job demands as well as job resources. For instance, home-office and teleworking can serve as a job resource and are associated with positive influences on employee health, such as better sleep and more physical activity [8], lower stress levels, and better overall physical health [9] and lower burnout values [10]. On the other hand, cognitive job demands in particular are supposed to rise with increasing digitalization [11,12].

Digital transformation processes are considered a prime challenge for leadership and top management [3,13,14]. Managers must drive digital transformation within the company, keep up to date with latest technological developments, and make difficult decisions with far-reaching consequences. Work has therefore changed, particularly for managers and is today often more cognitively complex, more collaborative, and more time pressured than it used to be [15]. This may especially be the case for upper-level managers of internationally operating companies with increasing global competition and high-level work responsibilities. Organizations are also more agile nowadays and processes are often unclear, fluid, and uncertain. As a result, managers often have many options available and many (complex) decisions, with adverse consequences, to make. The perceived choice overload is the result of a complex interaction between psychological processes, including awareness of opportunity costs as well as regret in the case of wrong decision-making [16,17]. Although digitalization and its implications on well-being are often matter for debate, the actual effects have not yet been well researched It can be assumed that the burden of leadership decisions increases in modern workplaces. There are more complex processes and a large amount and/or ambiguous information that needs to be processed. We assumed that managers perceive high cognitive job demands during the digital transformation processes. For the purpose of our study we focused on two specific cognitive job demands: choice overload and pressure from digitalization. Both concepts have been operationalized into three items. Choice overload was assessed by the (1) burden of leadership decisions, (2) agony of choice, and (3) complexity of decisions. Pressure from digitalization was defined as the pressure (1) to keep up with the latest technology, (2) to prepare for digitalization and find adequate answers, and (3) to keep up to date with the latest technological developments.

The current study analyzes the associations of both job demands with psychological well-being in upper-level managers. Our research question was as follows: Do choice overload and pressure from digitalization affect the probability that managers have low psychological well-being and if so, how strong is the influence?

We assumed that these demands may be perceived as a burden or strain and may negatively influence managers’ psychological well-being. The underlying hypotheses are: 

**Hypothesis 1** **(H1).***The higher the perceived ‘choice overload’, the higher is the probability for upper-level managers of having a low psychological well-being*. 

**Hypothesis 2** **(H2).***The higher the perceived ‘pressure from digitalization’, the higher is the probability for upper-level managers of having a low psychological well-being*. 

Our first hypothesis goes along with the choice overload hypothesis, which states that too many and too complex choices cause adverse consequences [17,18,19]. Multiple studies support the validity of this construct [18,19,20]. However, to our knowledge, there are so far no investigations of the choice overload hypothesis in managers and no studies that have tested associations of this construct with managers’ well-being. Since we could not find validated scales to assess our constructs of interest in managers, we developed two new scales based on a qualitative pilot study and previous research and concepts.

## 2. Theoretical Background

The broader theoretical framework for our study is provided by the Job Demands-Resources (JD-R) model from Bakker and Demerouti [21]. The model is one of the most widely studied models of occupational stress and proposes that working conditions can be categorized into two broader categories: job demands and job resources [21]. Excessive job demands, when not accompanied with adequate job resources, have been associated with reduced health and higher risks of burn-out [22,23]. A second useful model for our study is the Job-Demand-Control (JDC) model, developed in 1979 by Karasek [24]. The JDC model states that high job control (i.e., decision authority or work autonomy) results in positive health outcomes. In contrast, high job demands, coupled with low job control, have been associated with outcomes relating to risk of mental health disorders and sickness absence. In general, managers are perceived to have high job demands and high job control [25].

### 2.1. Choice Overload in Managers

The theory of choice overload, in accordance with the JDC model, states that many options (e.g., high job control) are health-promoting, but in contrast with the JDC model, it also states that beyond a certain level, too many options create excessive demand [17,26,27]. Schwartz (2005) states that ‘the fact that some choice is good doesn’t necessarily mean that more choice is better […] there is a cost to having an overload of choice’ [28]. Lehner et al. (2013) found that when the perceived scope of latitude is too large, the wish for decision latitude reduction occurs [16]. Further studies support the idea of the negative effects of choice overload [19,29]. The term choice overload is typically used in relation to a scenario where the complexity of an individual’s decision problem exceeds his or her cognitive resources [30,31]. A concept developed by Pfaff (2013) refers to the stress that results from having too many options [27]. The concept suggests that choice overload is experienced ‘if a person appraises a situation, which is characterized by many or difficult choices, as taxing exceeding his or her resources and endangering his or her well-being’ [27]. Choice overload can be described as the result of imbalances between demands and resources and can also be defined as a situation in which, from the point of view of the person concerned, the personal, social, and organizational resources are insufficient to adequately handle the quantity and/or quality of options [27]. Thus, choice overload is an individual demand that results from a psychological burden. This burden occurs from an overload of options and is a ‘result of a complex interaction between psychological processes including awareness of opportunity costs, rising expectations, an aversion of trade-offs, as well as regret, and self-blame in the case of wrong decision making’ [16]. Furthermore, according to the model of Pfaff (2013), a distinction can be made between quantitative and qualitative choice overload. Quantitative choice overload can occur when too many options are available. Qualitative overload can occur if a person has to make complex and difficult decisions [26,27]. Choice overload is a concept that has not yet been widely explored, especially in the field of stress research and work and organizational psychology. However, there is research on the phenomena of choice overload, but this exists primarily in consumer research—that is, how having many options affects purchase decisions among consumers. According to a meta-analysis by Chernev et al. (2015), four key factors moderate the effect of choice overload in consumer choice: choice set complexity, decision task difficulty, preference uncertainty, and decision goal [30]. However, research about how choice overload and specifically how a high complexity in choice overload, impacts managerial health is, to our knowledge, new. Because managers are potential role models and might serve as multipliers, we considered this sample as extraordinarily interesting.

### 2.2. Psychological Well-Being in Managers

Work and working conditions are important determinants of psychological well-being [32,33,34,35]. Studies have shown that high job demands and adverse psychosocial factors are significantly associated with poor well-being [33,36,37,38]. Poor psychological well-being is a signal of distress and an indication of possible depression. Reduced psychological well-being affects the individual’s health and can, in the long-term, lead to depression, productivity lost, and absenteeism [36,38]. Managers are exposed to high levels of work demands and previous studies have shown that managers already experience high degrees of distress [39,40,41]. In a study by Fiedler et al. (2018), 25.4% of the surveyed managers were classified as having poor well-being, indicating that these managers are at risk of developing depression. In the same study, 10.3% of the managers already reported symptoms of depression [39]. A study by Nyberg et al. (2009) showed that distress can also affect the health (e.g., ischemic heart disease) of subordinates [42]. Further studies reported that managers play a crucial role for the health and well-being of employees [43,44,45]. Improving managers’ well-being is therefore in the best interest of organizations and communities. 

## 3. Materials and Methods 

This study was part of a broader project on ‘digitalization and health’. A survey of upper-level managers from a large German ICT-company was conducted to: (1) identify specific job demands among upper-level managers in the context of digitalization and (2) ask the managers about their view on the current status of digitalization and health. At the time of the study, extensive transformation processes were taking place in the company (e.g., reorganizations, implementation of new technologies). As specific job demands in the context of these digital transformation processes, choice overload and perceived pressure from digitalization were particularly of interest in the current study. Both constructs were named as specific demands in a previous pilot study. Five expert interviews were conducted before developing the survey to obtain views on (1) the current digital transformation in the company and (2) implications for managers’ workload and well-being. Participation in the study was voluntary. The data were anonymously collected and analyzed. Participants gave their consent for the survey and permission to analyze all information from the questionnaire and for publication in anonymized form for research purposes by the University of Cologne. The project was supported by two cooperating project partners (a German health insurance company and a German company in the ICT-sector). For reasons of data security, the project partners and funders of the study want to remain anonymous.

### 3.1. Study Design and Participants

The cross-sectional study lasted from June to July 2017. The data were obtained using a web-based survey tool. All upper-level managers (*n* = 1760) of the company were invited to participate in the survey. Upper-level managers, in this context, are executives responsible for managers in lower management. Our study participants have high responsibilities and must often define what kind of goal should be achieved, rather than find the right means to achieve this goal. The company’s Chief Human Resources Officer (CHRO) supported the study and encouraged the managers to participate in the survey. He informed the participants that their privacy would be protected, explained the procedure, and highlighted the possible benefits of the survey for the managers and for the company’s future digitalization strategy. The study design and realization were presented to the Ethics committee of the University of Cologne, Medical Faculty (application No. 18-208) No objections to any aspects of the study were raised.

The survey was designed according to Dillman’s Total Design Method [46]. Four e-mails were sent out by the company’s CHRO. The first e-mail was to notify participants before the start of the survey and the second e-mail was sent out at the beginning of the survey. Two reminders at one-week intervals were sent out to increase the response rate. To reach all executives, the questionnaire was available in English and German. The selection of the sample is presented in Figure 1. A total of 1760 upper-level managers were informed about the survey by the company’s CHRO. Of the 1760 managers, 2 managers sent an e-mail stating that they no longer performed any leadership tasks. Of the remaining 1758 managers, 8 sent an e-mail refusing to participate for various reasons. Of the remainder, 1175 managers did not participate in the survey and did not respond to any of the e-mails. Ultimately, 575 managers completed the questionnaire. Data from 207 participants who filled in less than 30% of the questionnaire were excluded. A total of 368 upper-level managers completed more than 70% of the survey questionnaires and were included in the analysis sample (response rate: 20.9% of the net sample). The managers’ characteristics are shown in Table 1.

### 3.2. Measures

Psychological well-being in managers was assessed using the English and German version of the World Health Organization Well-Being-Index (WHO-5) [47,48]. For choice overload and pressure from digitalization, two new scales were developed in this study. As far as we know, at the time of the study, there were no comparable questionnaires measuring choice overload or pressure from digitalization in managers. The questionnaire was developed and pretested in German. Cognitive interviews were conducted to ensure the survey met the purpose of our study, to avoid problems with comprehension and to test for face validity. Afterwards the scales were discussed and refined by a multidisciplinary team of six experts (see Section 3.2.2). The questions were then translated into English by a native speaker. To control for differences in manager characteristics, we included data on age, gender, years of managerial experience, and managerial responsibility.

#### 3.2.1. Dependent Variable

Psychological well-being is a multidimensional concept that includes aspects of self-esteem and satisfaction with life. The construct refers to having a positive view of one’s self and one’s life and occurs when there is an absence of mental disorders and a presence of positive states [49,50]. The World Health Organization (WHO) defines mental well-being as ‘a state of well-being in which the individual realizes his or her own abilities, can cope with the normal stresses of life, can work productively and fruitfully, and is able to make a contribution to his or her community’ [51]. The WHO-5 is a positively worded, self-administered questionnaire measuring psychological well-being within the previous two weeks. It is a widely used instrument and is considered a valid tool for measuring positive psychological well-being [52]. The questionnaire has been translated into more than 30 languages and has been used in research all over the world [39,52,53,54,55]. The WHO-5 covers five items related to positive mood (good spirits, relaxation), vitality (being active and waking up fresh and rested) and general interests (being interested in things). In the present study, internal consistency was α = 0.87. The surveyed managers responded on a 6-point Likert scale ranging from 0 ‘not present’ to 5 ‘constantly present’. The total scale score, ranging from 0 to 25, was then calculated for each manager by summing up the five item scores. The minimum possible score 0 indicated the worst possible state of well-being, while the maximum possible score of 25 indicated the best possible state of well-being. The WHO-5 has been shown to be a sensitive and specific screening tool for depression. Cut-off scores indicating poor or high psychological well-being are well established [55]. A raw score below 13 indicates poor well-being and is an indication for testing for depression under ICD-10 [56].

#### 3.2.2. Independent Variables

As far as we know, there are no suitable and validated questionnaires for our two independent variables. Based on a qualitative pilot study and a detailed literature review, new scales for the constructs of interest were developed. The newly developed scale for choice overload was based on the choice overload hypothesis and concept developed by Pfaff (see Section 2.1). From our point of view, choice overload is especially relevant for managers who must make important decisions. The scale we developed has been adapted accordingly and consists of three items measuring the following: (1) burden of leadership decisions, (2) agony of choice, and (3) complexity of decisions. For the present study, internal consistency had a Cronbach’s α = 0.71. Based on the literature and our qualitative pilot study, we also assumed that the pressure on upper-level managers in the course of digitalization increases. As a second cognitive demand, we wanted to capture the pressure that comes with the current digital transition. We developed a new scale with three items assessing the following types of pressure: (1) to keep up with the latest technology, (2) to prepare for digitalization and find adequate answers, and (3) to keep up to date with the latest technological developments. For the present study, internal consistency was Cronbach’s α = 0.88. 

For both scales, each of the items had to be answered on a four-point Likert scale ranging from 1 ‘disagree completely’ to 4 ‘agree completely’. All items were discussed and refined by a team of six experts from different occupations (including an IT-specialist, occupational health specialists, and specialists in health questionnaire development). After development, the scales were pretested in our target group, according to the think-aloud method. The questionnaire was then translated into English by a native speaker.

#### 3.2.3. Confounding Variables

Several variables were included in the analysis to control for their potential confounding effects on our dependent variable. Because well-being has been shown in previous research to be a complex and multidimensional concept with differences in certain variables (e.g., gender and socioeconomic status) [57], the following variables were thought to possibly be related to our dependent variable and were included in the analysis: gender, age, managerial experience and managerial responsibility. Gender was dichotomized as male or female. Age was measured in five categories (<30 years; 31–40 years; 41–50 years; 51–55 years; >55 years). Managerial experience was assessed in full years. Managerial responsibility was measured by the following six categories based on the number of employees for whom the respondent was responsible for: 1–9, 10–99, 100–999, 1000–4999, 5000–9999, 10000 or more. 

#### 3.2.4. Statistical Analysis

Responses for well-being were scored and dichotomized into groups of high and low well-being according to the well-established cut-off score of <13 for the WHO-5 [55]. To assess possible statistical differences between the group of managers with low well-being and the group with high well-being, chi-square, and t-tests were conducted (see Table 2).

Stepwise multivariate logistic regression analysis was performed to examine the relationships between well-being and the independent variables (see Figure 2). Years of managerial experience, choice overload, and pressure from digitalization were used as continuous variables. The variables gender, age, and managerial responsibility were used as categorical variables. In Model 1 of the stepwise multivariate logistic regression analysis, we tested the unadjusted effects of all variables (crude analysis). In the following models, we tested the effects of the two independent variables adjusted for confounding variables (Models 2–3). Odds ratios (OR), p-value, corresponding 95% confidence intervals (CI) and Nagelkerke’s pseudo-R2 were calculated. In Model 4, we tested the full model with both independent variables, adjusted for the confounding variables. No missing values were imputed. Data from managers who did not finish at least 70% of the questionnaire were excluded from the analysis. All statistical analyses were performed using SPSS 25 (IBM, Armonk, NY, USA) for Windows. A *p* value of less than 0.05 was considered statistically significant.

## 4. Results

### 4.1. Sociodemographic Characteristics of the Participants

A total of 368 managers participated in the study. Of these, 14 managers (3.8%) filled in the English version of the questionnaire. Table 1 presents the sociodemographic characteristics of the study participants. Of the 368 participants, 76.9% were male and 23.1% were female. The average managerial experience was 11.5 years, with a standard deviation of 6.73. Managerial experience had a range from 0 to 50 years. The average score for psychological well-being was 15.73, with a standard deviation of 4.6 and a range from 0 to 25. The findings show that 21.5% of the surveyed managers were classified with poor well-being (*n* = 72) and 78.5% with high well-being (*n* = 263). The average of perceived choice overload and pressure from digitalization among all participants was medium to high (M = 7.62, SD = 1.89; M = 8.23, SD = 2.31). Bivariate comparisons between the low and high well-being groups using a t-test revealed a *p* value <0.001 for choice overload and a *p* value > 0.05 for pressure from digitalization and managerial experience (see Table 2). In a chi-square test we found p values < 0.001 for the variables gender, age and managerial responsibility (see Table 2).

### 4.2. Association between Job Demands and Psychological Well-Being

Table 3 summarizes the associations between psychological well-being and the independent variables based on stepwise multivariate logistic regression analysis. Model 1 of the multivariate analysis shows the unadjusted model (crude analysis). Models 2–3 show the models for the two independent variables adjusted for the covariates gender, age, managerial experience, and responsibility. Model 4 shows the full model with both independent variables and adjusted for confounding variables (see Table 3). Nagelkerke’s R square is 0.102 for the full model (Model 4), which, according to Cohen (1992) corresponds to a moderate effect [58]. The results show that a higher degree of perceived choice overload was significantly associated with low psychological well-being (*p* < 0.01, OR = 1.246). Perceived pressure from digitalization showed no significant association with psychological well-being in our sample. Our findings also show that gender, age, managerial experience and managerial responsibility have no effect in our model (see Table 3).

## 5. Discussion

The present study examines whether there is an association between subjectively perceived choice overload, pressure from digitalization, and psychological well-being in a sample of upper-level managers. Since we could not find validated scales for both constructs at the time of our study, we developed two new scales for our independent variables. The development was based on a qualitative pilot study and previous research on job demands among managers. Our findings provide evidence of an association between perceived choice overload and psychological well-being. In line with our expectations, we found that managers with high choice overload are more likely to have low psychological well-being. We have therefore found evidence to support our hypothesis that choice overload affects the probability that managers have low psychological well-being (hypothesis 1). The results are—to our knowledge—the first to support this hypothesis among upper-level managers. Against our expectations, we found that the subjectively perceived pressure from digitalization had no effect on well-being in our sample (hypothesis 2). One explanation might be that our study took place in an ICT-company and that the participating managers are considered IT-experts. An assumption would be that IT-experts have a technological affinity and feel the pressure from digitalization, but do not perceive this as a burden. This goes along with the JDC model, which states that job demands are associated with reduced health outcomes if a person does not have sufficient resources to balance the demands. As studies have shown that managers typically have high work requirements, we can assume that high pressure from digitalization might only be associated with a lower level of well-being when accompanied with low job resources. Since we didn’t consider job resources in our study, this aspect should be tested in further studies. If this assumption is confirmed, a practical implication would be that job resources are a relevant starting point to buffer high job demands and for improving the well-being of managers. 

Our results have further shown that approximately every fifth manager (21.5%) was categorized with low psychological well-being. Because previous studies have shown that a score for well-being below 13 is a first indication for depression [55,59] and that managers also have an influence on employee well-being, we perceive this result as an urgent call to action. However, the findings of our study are comparable to results from other studies using the WHO-5. For instance, the Fifth European Working Conditions Survey has shown that 19.5% of managers across the EU report a score below 13 for psychological well-being [60]. In addition, a study by Fiedler et al. has shown that 25% of surveyed managers were classified as having low psychological well-being [39]. These quite high levels of low well-being support the growing need for attention to be paid to psychological well-being in managers, especially in times of digital transition.

As the scales for our independent variables were newly developed for this study, further verification and validation of the scales is necessary. In the present study internal consistency was good for pressure from digitalization (α = 0.88) and acceptable for choice overload (α = 0.71). One reason why the internal consistency of the scale for choice overload might not be in the good range could be that choice overload is a complex construct with several aspects (i.e., qualitative and quantitative) and cannot be adequately covered in three items. However, due to space restrictions, we unfortunately had to shorten our questionnaire, which might have led to limitations in the study scales. Nevertheless, our findings show that choice overload seems to be a cognitive job demand important for managers’ well-being. Furthermore, the presumption of reverse causation should be discussed: it could also be assumed that those managers with lower psychological well-being may be experiencing more choice overload due to impaired cognitive function. The present study could be used as a starting point for further examination of this assumption. Further research is needed to confirm the direction of the relationship between choice overload and psychological well-being. 

In addition, the possibility that the investigated job demands may be non-linearly associated with well-being, should be discussed. This assumption goes along with the Vitamin Model of Peter Warr (1987), which hypothesized that three job characteristics (i.e., job demands, job autonomy, and workplace social support) are curvilinearly related with key indicators of well-being [61]. Further research on choice overload and pressure from digitalization could investigate this hypothesis.

In our study, we collected data only in one company. The study sample might therefore not be representative, and we must assume that this limits the generalizability of our results. Furthermore, the low response rate (20.9%) might also have biased the results. Reasons for the low response rate may have been that the survey took place during the summer period, as well as the relatively short run time of our study (4 weeks). Another reason might be that (upper-level) managers in general are a very busy and hard-to-reach target group.

## 6. Limitations and Further Study

### 6.1. Strength and Limitations

Several limitations to our study should be mentioned. First, all observations were based on self-reports, which can cause effects to be over- or underestimated [62]. However, our constructs of interest are mental constructs, which only can be indirectly observed. Second, our analysis sample consisted of 20.9% of the company’s upper-level managers; the relatively low response rate and the single-site company data collection strategy may have biased our results and limits the generalizability. Third, the data for our study were obtained using a cross-sectional study design. Developments over time and causal conclusions are not appropriate on the basis of cross-sectional data [63]. Furthermore, the scales for choice overload and pressure from digitalization were newly developed for our study and have not yet been scientifically validated. Testing concurrent validity was also not possible, because there are no comparable questionnaires. The pressure from digitalization scale has shown good internal consistency (α = 0.88), while for the choice overload scale, internal consistency was α = 0.71, which is acceptable, but shows that the scale could still be improved. We assume that the choice overload scale could be strengthened by measuring other indicators such as decision confidence, satisfaction with choices and regret. Future studies could also include the behavioral consequences of choice overload, including the likelihood of deferring the choice or reversing an already-made choice. Furthermore, in our research model, we included information on gender, age, managerial experience and managerial responsibility, but we can’t exclude the possibility that further factors might have influenced our results.

### 6.2. Implications for Practice and Further Research

Working conditions are changing, and perceived stress has risen and can lead to mental diseases. Our study provides the first evidence that choice overload is negatively associated with psychological well-being in managers. Further research is needed to test the generalizability of our results and to confirm the direction of the relationship between choice overload and psychological well-being. If further research investigates whether improving choice overload improves well-being, choice overload should be considered in labor organizations for managers to ensure their long-term health. Because decision-making processes are particularly important for managers in digital transformation processes, we believe that the reduction of choice overload could be a possible approach to the maintenance of managers’ health in the digital era. Due to the nature of the job profile, if the magnitude of choice overload cannot be changed, then at least conveyed coping strategies in the form of stress management training could improve their self-empowerment by strengthening their internal resources. One possible implication might be to develop and test training and education to improve such coping strategies in future research. Both individual as well as collective coping strategies are available: special techniques can help to focus attention while suppressing irrelevant options, along with the use of disburdening heuristics on the basis of employee experiences. Some of the decisions to be made can also be standardized and supported by predefined procedures. Given these implications, it is important to further understand the conditions under which the adverse effects of choice overload and pressure from digitalization are likely to occur. Interviews may provide further guidance on how to tackle the issues of cognitive demands in the occupational setting for upper-level managers.

## 7. Conclusions

With increasing digitalization, the manager’s job complexity continues to increase. This study provides preliminary insight into the relationship of two cognitive job demands—choice overload and pressure from digitalization—and psychological well-being among upper-level managers. Both concepts have not yet been well researched. The findings of our study serve as a starting point for further investigations on the phenomena and occupational health-related outcomes. Although the topic requires further study, managers’ cognitive job demands should be considered in the practice of health promotion because of their possible negative effect on managers’ health. Given the unsettled state of the field, it is important to try to further understand when choice overload and pressure from digitalization occur and when these may trigger negative health consequences. In previous studies, the focus has tended to be on employees (i.e., implications of digitalization on employee work and health) and there have been few studies on the implications of digitalization for managers, particularly in upper management. Managers are, however, an important target group because they have a proven impact on the company and on their subordinates.

## Figures and Tables

**Figure 1 ijerph-16-01746-f001:**
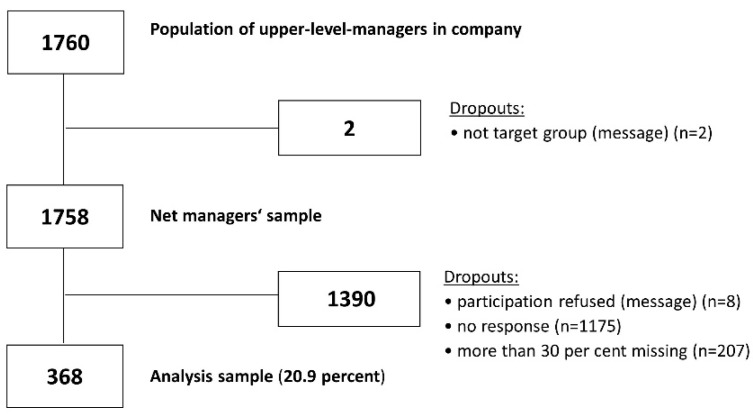
Flowchart of the selection of the managers’ sample.

**Figure 2 ijerph-16-01746-f002:**
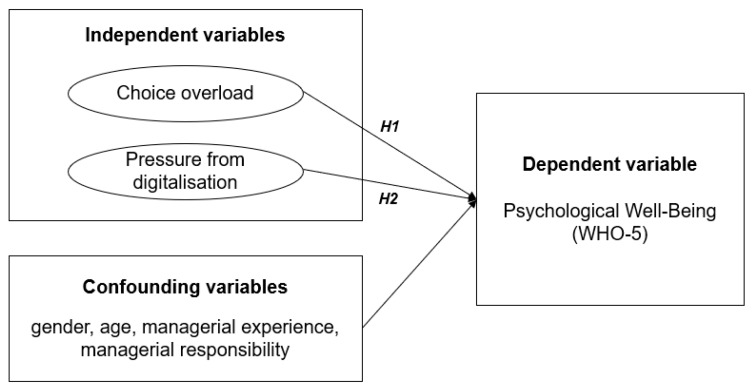
Research model for statistical analysis.

**Table 1 ijerph-16-01746-t001:** Descriptive characteristics for all model variables.

**Variable**	***n***	**M**	**SD**	**Median**	**Minimum**	**Maximum**
Choice overload	368	7.62	1.89	8	3	12
Pressure from digitalization	368	8.23	2.31	9	3	12
WHO-5	335	15,73	4,60	16	0	25
Managerial experience	334	11,50	6,73	10	0	50
		**Response Trait**		**Frequency (*n*)**		**Percentage**
WHO-5 (dichotomized)	335	Low (raw value <13)		72		21.5
		High (raw value ≥13)		263		78.5
Gender	334	Male		257		76.9
		Female		77		23.1
Age	334	<30		0		0.0
		41–50		157		47.0
		51–55		126		37.7
		>55		38		11.4
Managerial responsibility	334	1–9		55		16.5
		10–99		191		57.2
		100–999		74		22.2
		1000–4999		11		3.3
		5000–9999		1		0.3
		10000 or more		2		0.6

**Table 2 ijerph-16-01746-t002:** Descriptive statistics of the independent variables for managers with high and low psychological well-being.

	Managers with High Well-Being				Managers with Low Well-Being				*t*-Test *p* Value
Variable	*n*	M	SD	Median	*n*	M	SD	Median	
Choice overload	263	7.46	1.87	7.0	72	8.22	1.94	8.0	<0.05
Pressure from digitalization	263	8.20	2.32	9.0	72	8.46	2.34	9.0	>0.05
Managerial experience	262	11.42	6.34	10.00	72	11.82	8.05	10.00	>0.05
**Variable Response Trait**	**Managers with High Well-Being Percentage**				**Managers with Low Well-Being Percentage**				**Chi-SquareTest** ***p* Value**
Gender									
Men		75.6				81.9			<0.001
Women		24.4				18.1			
Age									
30		0				0			<0.001
31–40		3.8				4.2			
41–50		48.5				41.7			
51–55		35.5				45.8			
>55		12.2				8.3			
Managerial responsibility									
1–9		14.9				22.2			<0.001
10–99		58.8				51.4			
100–999		21.8				23.6			
1000–4999		3.8				1.4			
5000–9999		0.4				0			
10000 or more		0.4				1.4			

**Table 3 ijerph-16-01746-t003:** Results of the stepwise Logistic Regression Analysis.

	Model 1 Crude Analysis			Model 2 Choice Overload, Adjusted			Model 3 Pressure from Digitalization, Adjusted			Model 4 Full Model, Adjusted		
Variable	OR	95% CI	*p*	OR	95% CI	*p*	OR	95% CI	*p*	OR	95% CI	*p*
Choice overload	1.235	1.074–1.421	0.003	1.230	1.065–1.421	0.005	-	-	-	1.246	1.064–1.459	0.006
Pressure from digitalization	1.050	0.937–1.175	0.401	-	-	-	1.050	0.935–1.179	0.412	0.975	0.857–1.109	0.701
Gender	1.467	0.756–2.848	0.258	1.583	0.791–3.168	0.194	1.592	0.802–3.161	0.184	1.583	0.791–3.169	0.195
Age												
31–40 (1)	1.600	0.337–7.593	0.554	2.397	0.441–13.032	0.312	2.092	0.392–11.164	0.388	2.327	0.424–12.763	0.331
41–50 (2)	1.260	0.483–3.285	0.637	1.388	0.488–3.953	0.539	1.461	0.520–4.106	0.472	1.372	0.481–3.915	0.554
51–55 (3)	1.892	0.726–4.933	0.192	1.863	0.682–5.088	0.225	2.001	0.743–5.388	0.170	1.864	0.682–5.091	0.225
>55 (reference)												
Managerial experience	1.009	0.971–1.048	0.652	1.014	0.969–1.062	0.538	1.013	0.968–1.059	0.583	1.014	0.968–1.061	0.564
Managerial responsibility												
1–9 (1)	0.410	0.024–6.967	0.538	0.661	0.037–11.805	0.778	0.523	0.029–9.294	0.659	0.647	0.036–11.606	0.768
10–99 (2)	0.240	0.015–3.931	0.317	0.374	0.0122–6.460	0.499	0.287	0.017–4.920	0.389	0.369	0.021–6.385	0.493
100–999 (3)	0.298	0.018–5.025	0.401	0.440	0.025–7.870	0.577	0.356	0.020–6.340	0.482	0.432	0.024–7.737	0.568
1000–4999 (4)	0.100	0.003–3.153	0.191	0.118	0.004–3.949	0.233	0.099	0.003–3.287	0.196	0.117	0.004–3.918	0.231
5000–9999 (5)	0.000	0.000	1.000	0.000	0.000	1.000	0.000	0.000	1.000	0.000	0.000	1.000
10000 or more (reference)												
Cox & Snell R Square						0.053			0.031			0.053
Nagelkerke’s pseudo-R Square						0.082			0.048			0.082

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
