# Peer review of "Managers’ Well-Being in the Digital Era: Is it Associated with Perceived Choice Overload and Pressure from Digitalization? An Exploratory Study"

_ijerph, 2019, doi:10.3390/ijerph16101746_

Reviewer 1 Report

This is a very interesting, original and timing study that is appropriate for publication in the Journal. I compliment the authors for their analyses and clarity in exposing their results. I have some minor concerns that the authors may elaborate in this study or in their future projects in the field. The sample size of this study may be strongly biased by the low response rate and by the single-site company data collection strategy. Authors reported this in the limitations part of the study, but may be more stressed out in the discussion part. Authors also based their hypotheses on the resource-demand and Karasek`s models, but they in fact "neglect" the resources aspect in their study design. Also, we don`t really know if work environmental factors, as well as personal factors, may have influenced the results. This is not mentioned in the limitation part of the study. Despite this, I find the study very well conducted and an important first step for further studies on the topic. I would like to invite the authors to consider also the Vitamin Model of Peter Warr for the development of their hypotheses; it may be that some factors don`t follow a linear pattern (e.g., the more the better or vice-versa). 

Author Response

Response to Reviewer 1 Comments

Point 1: The sample size of this study may be strongly biased by the low response rate and by the single-site company data collection strategy. Authors reported this in the limitations part of the study but may be more stressed out in the discussion part.

Response 1:

We are very grateful to reviewer 1 for taking the time to make constructive criticism on how this manuscript could be improved. We agree with the comment and have now focused on this aspect in more detail. We now wrote:

“In our study, we collected data only in one company. The study sample might therefore not be representative, and we must assume that this limits the generalizability of our results. Furthermore, the low response rate (20.9%) might also have biased the results. Reasons for the low response rate may have been that the survey took place during the summer period, as well as the relatively short run time of our study (4 weeks). Another reason might be that (upper-level) managers in general are a very busy and hard-to-reach target group.” (see p.12, lines 356-361)

In addition, we added in our implication chapter and limitation chapter:

“Further research is needed to test the generalizability of our results and to confirm the direction of the relationship between choice overload and psychological well-being.” (see p.12, lines 378-380)

 “Second, our analysis sample consisted of 20.9% of the company’s upper-level managers; the relatively low response rate and the single-site company data collection strategy may have biased our results and limits the generalizability.”       (see p.12, lines 365-368)

Point 2: Authors also based their hypotheses on the resource-demand and Karasek`s models, but they in fact "neglect" the resources aspect in their study design.

Response 2:

That’s right, in our study we focused only on job demands. We stated this in our introduction and derived implications on how job resources might influence the association in the discussion part. We now extended the discussion of this aspect:

“As studies have shown that managers typically have high work requirements, we can assume that high pressure from digitalisation might only be associated with a lower level of well-being when accompanied with low job resources. Since we didn’t consider job resources in our study, this aspect should be tested in further studies. If this assumption is confirmed, a practical implication would be that job resources are a relevant starting point to buffer high job demands and for improving the well-being of managers.” (see p.11, lines 321-326)

Point 3: Also, we don`t really know if work environmental factors, as well as personal factors, may have influenced the results. This is not mentioned in the limitation part of the study.

Response 3:

In our study, we added information on gender, age, managerial experience and managerial responsibility of the study participants. We totally agree that further work environmental and personal factors may have influenced our results. We therefore added this aspect in our limitation part. (see p.12, lines 381-383)

“Furthermore, in our research model, we included information on gender, age, managerial experience and managerial responsibility, but we can’t exclude the possibility that further factors might have influenced our results.”

Point 4: I would like to invite the authors to consider also the Vitamin Model of Peter Warr for the development of their hypotheses; it may be that some factors don`t follow a linear pattern (e.g., the more the better or vice-versa). 

Response 4: Thanks for this helpful comment. The vitamin model works very well with our study. In the discussion part we added a paragraph about the model and the probability that some factors might not follow a linear pattern. (see p.11, lines 350-354)

“In addition, the possibility that the investigated job demands may be non-linear associated with well-being, should be discussed. This assumption goes along with the Vitamin Model of Peter Warr (1987), which hypothesizes that three job characteristics (i.e. job demands, job autonomy and workplace social support) are curvilinearly related with key indicators of well-being. Further research on choice overload and pressure from digitalisation could investigate this hypothesis.”

Reviewer 2 Report

Managerial responsibility was measured by the following six categories: 1–9, 10–99, 100–999, 1000–243 4999, 5000–9999, 10000 or more.- but there is no unit given? 

I have some doubts haw  Pressure from digitalisation is recognized? Haw it was measure what questions were asked - it could be more clearly desribed.

Author Response

Response to Reviewer 2 Comments

Point 1: Managerial responsibility was measured by the following six categories: 1–9, 10–99, 100–999, 1000–243 4999, 5000–9999, 10000 or more.- but there is no unit given?

Response 1:

Thank you very much for the comment. We agree and added additional information on the unit. We now wrote:

“Managerial responsibility was measured by the following six categories based on the number of employees for whom the respondent was responsible for: 1–9, 10–99, 100–999, 1000–4999, 5000–9999, 10000 or more.” (see p. 7. lines 249-251)

Point 2: I have some doubts haw  Pressure from digitalisation is recognized? Haw it was measure what questions were asked - it could be more clearly desribed.

Response 2:

Thank you very much for the advice. On page 6, lines 231-234 we wrote:

“We developed a new scale with three items assessing the following types of pressure: 1) to keep up with the latest technology, 2) to prepare for digitalisation and find adequate answers and 3) to keep up to date with the latest technological developments. For the present study, internal consistency was Cronbach’s α=0.88.”

To make more clear, how our independent variables were defined and measured, we added additional information for both constructs in our introduction chapter.

“For the purpose of our study we focused on two specific cognitive job demands: choice overload and pressure from digitalisation. Both concepts have been operationalized into three items. Choice overload was assessed by 1) burden of leadership decisions, 2) agony of choice and 3) complexity of decisions. Pressure from digitalisation was defined as the pressure 1) to keep up with the latest technology, 2) to prepare for digitalisation and find adequate answers and 3) to keep up to date with the latest technological developments.” (see p.2, lines 62-68)